# Establishment of an Improved ELONA Method for Detecting Fumonisin B_1_ Based on Aptamers and Hemin-CDs Conjugates

**DOI:** 10.3390/s22176714

**Published:** 2022-09-05

**Authors:** Xinyue Zhao, Jiale Gao, Yuzhu Song, Jinyang Zhang, Qinqin Han

**Affiliations:** Faculty of Life Science and Technology, Kunming University of Science and Technology, Kunming 650500, China

**Keywords:** fumonisin B_1_, aptamer, hemin-CDs, ELONA, corn flour

## Abstract

Fumonisin B_1_ (FB_1_) is a strong mycotoxin that is ubiquitous in agricultural products. The establishment of rapid detection methods is an important means to prevent and control FB_1_ contamination. In this study, an improved enzyme-linked oligonucleotide assay (ELONA) method was designed and tested to detect the contents of FB_1_ in maize (corn) samples. F10 modified with biotin was bound to an enzyme label plate that was coated with streptavidin (SA) in advance, and carbon dots (CDs) were used to catalyze the color of tetramethylbenzidine (TMB). The complementary chain of F10 was modified with an amino group and coupled with CDs to obtain conjugates. The sample and conjugates were then added to the enzyme plate coated with F10 (an FB_1_ aptamer). Upon completion of the color reaction, the absorbance was measured at 450 nm. The LOD of this method was 4.30 ng/mL and the LOQ was 13.03 ng/mL. We observed a linear relationship in the FB_1_ concentration range of 0–100 ng/mL. The standard curve was y = −0.001482 × x + 0.3463, R^2^ = 0.9918, and the experimental results could be directly measured visually. The recovery of the maize sample was 97.5–99.23% and 94.54–99.25%, and the total detection time was 1 h.

## 1. Introduction

Fumonisins (FB) are important members of the mycotoxin family which are produced by Fusarium moniliforme Sheld. The group is partly comprised of fumonisin A_1_ (FA_1_), fumonisin A_2_ (FA_2_), fumonisin B_1_ (FB_1_), fumonisin B_2_ (FB_2_), and fumonisin B_3_ (FB_3_), of which FB_1_ is both the most common and most toxic [1]. The European Union (EU) and US Food and Drug Administration (FDA) have set the upper limit of FB_1_ in feed to be 200–2000 μg/kg and 3000–4000 μg/kg, respectively [2]. Iran, France, Bulgaria, Switzerland, and other countries stipulate that the upper limit of FB_1_ contamination for corn-based breakfast cereals should be 800 μg/kg, while that for baby food should be estimated at 200 μg/kg [3]. Maize (corn) is the crop most polluted by FB_1_, though the toxin has also been found in wheat and soybeans. As maize is the most common crop used for producing animal feed, its contamination with toxins is likely to lead to serious adverse reactions in the animals consuming it [2]. The toxic mechanism of FB_1_ involves the inhibition of sphingolipid biosynthesis; an important cellular regulator; thus leading to organ failure [4,5]. Existing studies have shown that a variety of diseases in mammals can be caused by FB_1_, such as equine white matter encephalomalacia [6], porcine pulmonary edema [7], mouse liver tumorigenesis [8], and acute kidney poisoning in goats [9]. FB_1_ can also cause adverse effects in poultry including sudden weight loss, shrinkage of the organs such as the spleen, acute myocarditis, liver function injury, and increased mortality [10,11,12].

Several methods have been developed for the detection of FB_1_. The main common methods for FB_1_ detection are ELISA, molecularly imprinted polymers (MIPs), chromatography, electrochemistry, and immunochromatography (ICA). Munawar proposed one such competitive method based on molecularly imprinted polymer nanoparticles (MINA). In the method, FB_1_ was fixed on a solid carrier for the preparation of nano-molecularly imprinted polymers (MIPs). The LOD for the method was in the concentration range of 10 pM–10 nM is 1.9 pM [4]. Another method was developed by Qian et al., whereby a Max column was combined with high-performance liquid chromatography-tandem mass spectrometry (UPLC-MS/MS) to determine the content of FB_1_ in dairy products. The LODs of FB_1_ and FB_2_ in combination with their hydrolytic metabolites were 0.03 μg/L and 0.1 μg/L, respectively [13]. However, disadvantages of the above two outlined technologies include their expensiveness, high consumption of human and material resources, a relatively long detection time, and the requirement of having results analyzed by professionals. The electrochemical method possessed a relatively high sensitivity, a low LOD, and the advantage of strong specificity for target recognition. Zheng et al. carried out a series of chemical modifications on gold nanoparticles and graphene oxide (GO) to amplify the electrochemical signal to a certain extent and provide an anchor location for the modification of the FB_1_ recognition probe. From this, they established an electrochemical detection sensor and the LOD was 10 pg/mL [14]. Compared to the other methods, this method performed with good practicability and stability. However, a series of cumbersome pretreatments of the samples is required before monitoring can be performed. Based on the antigen-antibody reaction and the color development characteristics of colloidal gold, immunochromatography assays (ICA) with colloidal gold as the medium have been widely used in the field of rapid detection for human health and food safety applications. Ren et al. developed a colloidal gold test strip for FB_1_ detection by immunochromatography with the LOD of 2.5 ng/mL. This was used for the rapid detection of FB_1_ in maize [15]. Because antibodies are expensive, the research and development of test strips require a lot of human capital and material resources. In conclusion, the above methods have the obvious advantage of all being low LOD values; however, the cost of these methods is high and it is difficult to achieve rapid detection in the field. Therefore, there exists the need for a new method that introduces raw materials with lower costs, creates strong affinity and specificity, and uses suitable aptamers.

Aptamer is a single-stranded DNA or RNA that has the ability to recognize different types of targets, including cells, proteins, and small molecules [16]. Due to their excellent affinity and specificity, aptamers are often considered comparable to antibodies and have advantages in field detection due to their chemical synthesis, ease of chemical modification, and exponential self-amplification [17]. These advantages allow aptamers to be used in diagnostics, therapeutics, drug delivery, environmental monitoring, and food safety [18,19]. Aptamers also play a key role in food safety detecting. An optical probe nanosensor based on multi-walled carbon nanotubes, reduced graphene quantum dots, and was developed for the specific detection of organophosphorus pesticide diazinon with a detection limit of 0.4 nM (0.1 μg/L) [20]. FAM was used to modify its specific aptamer and TAMRA burst group was used to modify its complementary DNA to detect aflatoxin M_1_ in milk. In the presence of AFM1, the formation of the AFM1/aptamer complex resulted in a structural switch of the aptamer. The change in the aptamer structure lead to the release of cDNA, resulting in a fluorescent signal with a detection line of 0.5 ng/mL [21].

Specific FB_1_ inducers (F10) with high affinity were successfully selected by phylogenetic evolution of exponentially enriched ligands (SELEX) [22]. Carbon dots (CDs) were first discovered by Xu et al. while separating a carbon nanotube suspension by electrophoretic purification in 2004 [23]. Carbon dots have stable optical properties, catalytic activity, biocompatibility, and low toxicity, and therefore have a wide range of applications in food safety [24,25,26]. In this study, the method selects an aptamer that has high specificity and selectivity for specific targets, innovatively modifies the complementary chain of aptamers, binds the complementary chain of aptamers to carbon dots (CDs) through the amidation reaction, and uses the catalytic ability of CDs. A method for detecting FB_1_ by the ELONA based on an aptamer complementary chain and a fluorescent CDs conjugate was established. The method not only possesses all the advantages of the traditional ELISA; such as being fast, convenient, highly sensitive, and having low requirements for a sample pretreatment; it also possesses the unique advantage of the low cost resulting from using aptamers. As shown in Figure 1, when the assay FB_1_ is present in the system, FB_1_ and the aptamer complementary chain-carbon dot coupling form a competitive relationship and bind to the aptamer, leading to a decrease in the efficiency of the carbon dot being bound to the enzyme plate and a weakening of the catalytic effect, resulting in a low measured value of OD450.

## 2. Materials and Methods

### 2.1. Reagents and Apparatuses

FB_1_ was purchased from Binzhi Biotechnology (Shanghai, China). Ochratoxin A (OTA) was purchased from Xiheng Biotechnology (Shanghai, China). Zearalenone (ZEN) was purchased from Fubo Biotechnology (Beijing, China). Aflatoxin B_1_, aflatoxin G_1_, aflatoxin G_2_, 1-(3-dimethylaminopropyl)-3-ethylcarbodiimide hydrochloride (EDC), and 4-morpholineethanesulfonic acid (MES) were purchased from the Sigma-Aldrich Corporation (USA). Bovine serum albumin (BSA) was purchased from Shanghai Baisai Biotechnology Co., Ltd. (Shanghai, China). N-Hydroxysuccinimide was purchased from Shanghai Hongshun Biotechnology Co., Ltd. (Shanghai, China). All reagents used in the experiment were of analytical grade. The corn flour samples were purchased from local markets. A 0.01 M of phosphate-buffered saline (PBS) (pH 7.4) was used for fluorescence measurements. The carbonate buffer solution was prepared to contain 35 mmol/L NaHCO_3_ and 15 mmol/L Na_2_CO_3_ with ddH_2_O, and the pH was adjusted to 9.6. The oligonucleotides used in the experiment were synthesized by Tsingke Biological Technology (Beijing, China). The sequence of the F10 oligonucleotide is as follows: 5′-bio-CGATCTGGATATTATTTTTGATACCCCTTTGGGGAGACAT-3′. The sequence of the F10-com oligonucleotide is as follows: 5′-NH_2_-6(CH_2_)-ATGTCTCCCCAAAGGGGTAT-3′. An Agilent G9800A spectrofluorometer was used for measuring spectra. Thermo BIOMATE 3S was used to detect the results of the ELISA kit.

### 2.2. Synthesis of Hemin-CDs Conjugates

The Hemin-CDs were synthesized by the one-step hydrothermal method [27]. Choline chloride, glycine, and heme were calculated to have a molar ratio of 1:1:1. First, an amount of 0.606 g of urea, 1.39 g of choline chloride, 0.75 g of glycine, and 0.02 g of heme chloride were mixed in 40 mL of ddH_2_O and calcined in the autoclave at 180 °C for 8 h. Then, the resulting mixture was removed when the autoclave had recovered to room temperature and then centrifuged at 10,000 rpm for 10 min. Finally, the supernatant was filtered with a 0.22 μm filter membrane to obtain hemin-CDs conjugates, which were stored at 4 °C. The 10 μL preparation of hemin-CDs conjugates was placed in the refrigerator at −80 °C for freezing overnight and then freeze-dried with a freeze dryer before being weighed. The hemin-CDs conjugate preparation was then diluted to 30 mg/mL with ddH_2_O before usage.

### 2.3. Coupling F10-com with Hemin-CDs

The 300 mM NHS and 150 mM EDC solutions were prepared with ddH_2_O. A volume of 300 μL of 300 mM NHS, 300 μL of 150 mM EDC, and 325 μL of hemin-CDs conjugates were added into the 5 mL tube, then vortexed and vibrated for 1 min to fully mix the components, and stood at room temperature for 30 min. A volume of 300 μL of 10 mM F10-com and 400 μL of MES buffer was added to the above centrifuge tube and incubated overnight on a shaking table at 37 °C 120 rpm. Optimization of the F10-com concentration: F10-com was diluted to 10 μM with ddH_2_O. A volume of 300 μL of F10-com and 400 μL of MES buffer with different concentrations were added to the carboxyl-activated hemin-CDs conjugates so that the final concentrations of F10-com in the system were 0, 0.615, 1.23, 1.85, 2.46, and 3.07 μM, respectively. A volume of 35 μL of F10-com-hemin-CDs conjugates containing different concentrations of F10-com and 65 μL PBS was added to each well on the bio-F10 coated enzyme label plates, then gently shaken and mixed, and incubated at 37 °C at 60 rpm in a shaking table for 1 h. The microplates were then washed with PBST buffer 3X. A volume of 100 μL of TMB substrate solution was added to each well and incubated in the dark at room temperature for 30 min. Then, 100 μL of 10% sulfuric acid solution was added to each well to terminate the reaction and the absorbance values of all groups were measured within 10 min at 450 nm.

### 2.4. Verification of the Successful Coupling between Hemin-CDs and F10-com

Fluorescence migration method: A volume of 40 μL 30 mg/mL of hemin-CDs and a volume of 40 μL of F10-com hemin-CDs conjugates were aliquoted into two 5 mL centrifuge tubes, respectively. The volumes were fixed with PBS to 3000 μL, then vortexed and vibrated for 2 min to fully mix the solution. The parameters of the fluorescence meter were then set to the excitation wavelength of 322 nm under the emission condition, and the fluorescence spectra of the solutions in the two centrifugal tubes were measured.

Nuclear magnetic resonance: For F10-com-Hemin-CDs coupling product sample preparation, 365 μL of NMR buffer (10 mM NaCl, 12.3 mM KCl, 2 mM KH_2_PO_4_, 10 mM Na_2_HPO_4_, and 0.01% Tween-20, H_2_O: D_2_O = 9:1) was added to 135 μL of F10-com-Hemin-CDs coupling product in a total volume of 500 μL. For F10-com+Hemin-CDs sample preparation, 2.5 μL of NMR buffer was added to 470.5 μL of 100 μM F10-com and 27 μL Hemin-CDs in a total volume of 500 μL. The ^1^H spectra were performed at 288 K on a Bruker 600 MHz NMR spectrometer equipped with an H/C/N ultra-low temperature probe at 25 °C, all sampled using the W5 water peak.

### 2.5. Specificity Verification

A volume of 35 μL of F10-com-hemin-CDs conjugates and 10 μL of toxin diluted to 1 mg/mL was added to the F10-coated microplates so that the resulting toxin concentrations in the AFB_1_, AFG_1_, AFG_2_, ZEN, OTA, and FB_1_ groups were 100 ng/mL. A volume of 10 μL of ddH_2_O was added to the blank group and all the above holes were filled with PBS to 100 μL. After incubation in the dark at room temperature for 30 min, the microplates were washed with PBST buffer 3X and left to stand for 1 min each time. A volume of 100 μL of TMB substrate solution was added to each well and incubated in the dark at room temperature for 30 min. Then, 100 μL of 10% sulfuric acid solution was added to each well to terminate the reaction and the absorbance values of all groups were measured within 10 min at 450 nm.

### 2.6. Detection of the FB_1_ Standard Solution

A volume of 35 μL of F10-com-hemin-CDs conjugates, 35 μL of PBS, and 30 μL of FB_1_ standard solution diluted with ddH_2_O were added to the F10 coated microplates so that the resulting concentration of FB_1_ in each well was 0, 200, 400, 600, 800, and 1000 ng/mL, respectively. After 30 min of incubation in the dark at room temperature, the microplates were washed with PBST buffer 3X for 1 min each time. A volume of 100 μL of TMB substrate solution was added to each well and incubated in the dark at room temperature for 30 min. Then, 100 μL of 10% sulfuric acid solution was added to each well to terminate the reaction and the absorbance values of all groups were measured within 10 min at 450 nm.

### 2.7. Analysis of FB_1_ in Corn Flour Samples

The corn flour used in the experiment was purchased from a local supermarket as an edible raw material and the laboratory test showed no background values. Different concentrations of FB_1_ were added to the corn flour samples. Because FB_1_ is a water-soluble toxin, ddH_2_O was used to extract the toxin. An amount of 1 g each of two corn flour samples, A and B, was weighed, dissolved by 5 mL ddH_2_O, and then centrifuged at 6000 rpm for 20 min, respectively. The obtained supernatant was passed through a 0.22 μm filter membrane on the night and stored in a refrigerator at 4 °C for backup. FB_1_ standard was added to the above-pretreated samples until the total volume was 30 μL and the concentrations of FB_1_ were 20, 80, and 200 ng/mL, respectively. The spiked 30 μL sample, 35 μL of the F10-com-hemin-CDs conjugates, and 35 μL of PBS were added into the F10 coated microplates and shaken lightly to mix the components evenly. After 30 min of incubation in the dark at room temperature, the microplates were washed with PBST buffer 3X for 1 min each time. A volume of 100 μL of TMB substrate solution was added to each well and incubated in the dark at room temperature for 30 min. Then, 100 μL of 10% sulfuric acid solution was added to each well to terminate the reaction and the absorbance values of all groups were measured within 10 min at 450 nm.

## 3. Results and Discussion

### 3.1. Characterization of Hemin-CDs

The morphological characteristics of the hemin-CDs conjugates were observed by transmission electron microscope (TEM). As shown in Figure 1, hemin-CDs are spherical structures with uniform dispersion and the average particle diameter is 3.64 nm(The size of CDs was mostly less than 10 nm).

### 3.2. Verification of F10-com-Hemin-CDs Coupling Products

Fluorescence migration method: the fluorescence spectra of hemin-CDs conjugates and F10-com hemin-CDs conjugates were compared. As shown in Figure 2A, the fluorescence spectra measured by hemin-CDs showed their peak value to be at 400 nm, while the peak of F10-com-hemin-CDs was at the wavelength of 412 nm. The peak of the latter conjugate possessed a 12 nm shift, thus demonstrating that hemin-CDs was successfully coupled with F10-com, rather than the two conjugates having simply been mixed together. Nuclear magnetic resonance: as seen in Figure 2B, the peak spectrum of F10-com-hemin-CDs clearly produced new peaks between 0–5 ppm compared with the uncoupled peak spectrum. This was accompanied by the disappearance of some peaks, thus showing that there were new bonds formed alongside the disappearance of others. This indicates the occurrence of carboxyl dehydroxylation between the conjugates, followed by amino dehydrogenation, thus completing the amidation reaction. The result was the successful preparation of F10-com-hemin-CDs conjugates.

### 3.3. Optimization of Experimental Conditions

#### 3.3.1. F10 Encapsulated on Enzyme Labeling Plate

Detailed steps in the Appendix A.

#### 3.3.2. Optimization of F10-com Concentration

In the set gradient, OD450 initially increased and then decreased (Figure 3A). The reason for this is likely that F10-com was added to excess, such that some F10-com were not coupled with hemin-CDs successfully, but were still combined with F10 in the detection step. In this scenario, the amount of hemin-CDs bound to F10 was less, resulting in the reduction of its catalytic ability and the reduction of catalytic TMB ability. Therefore, the measured OD450 decreased. When the F10 com concentration was 1.85 μM, the value of OD450 was the greatest. Hence, the optimal concentration of F10-com was 1.85 μM. The fluorescent carbon dots prepared in this study were prepared by a one-step hydrothermal method and heme chloride was added to them to give the carbon dots good oxidation properties and can oxidize TMB, which is used as a catalyst instead of HRP in this method [28].

#### 3.3.3. Optimization of F10-com-Hemin-CDs Addition

To maximize the competition between hemin-CDs and FB_1_, it is necessary to saturate the concentration of F10-com in the system without the target FB_1_, so that hemin-CDs can exert their maximum catalytic ability. The F10-com-hemin-CDs conjugate preparation was added to each well on a bio-F10 coated enzyme label plate. The amounts of F10-com-hemin-CDs conjugates added were 0, 5, 10, 15, 20, 25, 30, 35, 40, 45, and 50 μL, respectively, within which the concentrations of F10-com were 0, 9.23, 18.46, 27.69, 36.92, 46.15, 55.38, 64.62, 73.85, 83.08, and 92.31 × 10^−3^ nM, respectively. These were supplemented to 100 μL with PBS and incubated at 37 °C 60 rpm for 1 h. The microplates were then washed with PBST buffer 3X. A volume of 100 μL of TMB substrate solution was added to each well and incubated in the dark at room temperature for 30 min. Then, 100 μL of 10% sulfuric acid solution was added to each well to terminate the reaction and the absorbance values of all groups were measured within 10 min at 450 nm. As shown in Figure 3B, with the increase of F10-com-hemin-CDs, the measured OD450 value increased gradually, and the color of the solution became darker, as assessed by the naked eye. Therefore, the F10 concentration of 64.62 × 10^−3^ nM was chosen as the best concentration.

### 3.4. Specificity Verification

To verify the selectivity of the method using the above-optimized conditions for each factor, the blank group, AFB_1_, AFG_1_, AFG_2_, ZEN, OTA, FB_1_, and toxin mixture were detected at the same time. Figure 4 displays the results measured when the concentration of toxin was 100 ng/mL. Compared with the blank control group, there was no obvious change in the measured results of AFB_1_, AFG_1_, AFG_2_, ZEN, and OTA groups. The OD450 of the FB_1_ group and toxin mixture group decreased significantly, and the color of the FB_1_ group and toxin mixture group was darker. This demonstrated that the method possessed high specificity.

### 3.5. Sensitivity of ELONA for FB_1_ Detection and Detection Range

Under the optimized conditions, the established detection method was used to detect the FB_1_ standard solution. The tested gradient concentrations of FB_1_ were 0, 200, 400, 600, 800, and 1000 ng/mL. The test result is shown in Figure 5A. When the concentration of FB_1_ was in the range of 0 ng/mL to 100 ng/mL, the absorbance value showed a good linear relationship with the concentration of FB_1_ and a linear relationship curve of y = −0.001482 × x + 0.3463 and R^2^ = 0.9918 was obtained. OD450 decreased gradually with the increase of the concentration of FB_1_, indicating that the catalytic ability of hemin-CDs decreased due to competition. Simultaneously, the color of the solution gradually became lighter, as observed by the naked eye. The calculation formula of the LOD was 3.3 × σ/S, where σ was the standard deviation of the test value of the blank group and S was the slope of the standard curve. Similarly, we used the equation 10 × σ/S to calculate the LOQ. Under the optimized conditions, the LOD value of the method was 4.30 ng/mL and the LOQ value was 13.03 ng/mL. FB_1_ and F10-com-Hemin-CDs were added to the enzyme plate precoated with F10, so the detection time of the method was 1 h.

By inferring from the standard curve, the maximum concentration of FB_1_ that can be detected by this method is 235.97 ng/mL. Through the detection of FB_1_ with different concentrations, when the concentration of FB_1_ was in the range of 0–200 ng/mL, there was still a good linear relationship between the FB_1_ concentration and OD450 (Figure 5B). However, when the concentration of FB_1_ exceeded 200 ng/mL, the value of OD450 plateaued. Therefore, the concentration range of FB_1_ detected by this method was 0–200 ng/mL. The detection range of the method in this study is 0–200 ng/mL, which has the advantage of being a wide detection range. If the concentration of FB_1_ in the food to be tested is high, the detection range of the method is small and false negatives may occur.

The detection process of the method in this study requires at least 1 h, which is relatively long compared with that of the electrochemical biosensor [29]. Therefore, more sensitive materials coupled with aptamers can be selected to establish a more rapid and sensitive detection method in future studies. In addition, electrochemical methods can be combined with competitive immunoreactivity and aptamer-based assays where different particles (e.g., AuNPs, magnetic nanoparticles, microplates) are functionalized with aptamers for fast detection, high specificity, and low LOD [30,31]. The improved ELONA method established in this study can be used for the subsequent development of commercial kits to detect FB_1_ in food and feed, like conventional ELISA kits after pre-treatment of enzyme-labeled plates with aptamer F10 and validation of its timeliness. The main problems of using current ELISA kits include high cost and narrow detection range, while the method established in this study can effectively circumvent these drawbacks.

FB_1_ is one of the more common mycotoxins in daily life, and mycotoxins usually do not appear alone, but usually there are multiple mycotoxins in moldy foods, and they are all very harmful to humans. Therefore, it is very important to establish a test method to detect multiple mycotoxins at the same time. Currently available methods that can simultaneously perform multiple toxin detections include immunochromatographic and electrochemical methods. Huang et al. established an immunochromatographic test strip based on gold nanoparticles (AuNPs), which can be applied to the simultaneous detection of FB_1_ and deoxynivalenol (DON) in traditional Chinese medicine with the detection limits of 20 ng/mL and 5.0 ng/mL for FB_1_ and DON, respectively [32]. Jiang et al. established a magnetron aptamer sensor for the simultaneous detection of OTA and FB_1_ using quantum dots (QDs)-coated silica as markers and complementary DNA-functionalized magnetic beads as capture probes, with detection limits of 0.10 ng/mL and 0.30 ng/mL for OTA and FB_1_, respectively [33].

### 3.6. Detection of FB_1_ in Corn Flour Samples

In order to verify the practicality of the method in real samples, different concentrations of FB_1_ were added to the pretreated samples. The standard addition and recovery experiments were carried out with the ELONA and ELISA kits constructed in this study, and the test results were compared. According to Table 1, the recovery rate of ‘corn flour sample 1′ measured by the method proposed by this study was 97.5–99.23%, and the recovery rate detected by the ELISA kit was 98.75–99.87%. The recovery of ‘corn flour sample 2′ by the method proposed by this study was 94.54–99.25%, and the recovery rate was 99.38–99.84% by the ELISA kit. Detection by the ELISA kit confirmed that the detection results of the method proposed by this study were accurate and sensitive and hence it can be used for the accurate monitoring of FB_1_ in food samples. The economic advantage of using aptamers in place of antibodies cannot be ignored, and the cost of using our method is significantly lower than that of the ELISA. Since the actual sample was tested with 1 g of corn sample dissolved in 5 mL of solution, the limit values of FB_1_ in corn flour of 800 μg/kg and 200 μg/kg [3] could be converted. With 200 μg/kg as the calibration value, 1 kg of corn flour was dissolved in 5 L solution, the upper limit of FB_1_ was converted to 40 ng/mL, and the LOD of this method was 4.30 ng/mL. Therefore, this method meets the detection requirements of the standard. The ELONA method pre-treatment is simple: ddH_2_O is used to dissolve the sample and extract FB_1_, and only simple weighing, dissolving, centrifuging, and filtering of the sample is required for the assay. LC-MS, on the other hand, requires more cumbersome sample handling methods [34].

## 4. Conclusions

To summarize, we successfully developed a method for the detection of FB_1_ by ELONA based on aptamer complementary chains and fluorescent carbon dot couples, using fluorescent carbon dots as catalysts for TMB color development. Under optimal conditions, the OD450 decreased linearly as the concentration of FB_1_ increased from 0 to 50 ng/mL, with an LOD of 13.03 ng/mL and a detection range of 0–200 ng/mL. The assay was highly specific for FB_1_ and did not detect other toxins with similar structures including AFB_1_, AFG_1_, AFG_2_, ZEN and OTA, and confirmed the feasibility of the method for use on maize flour samples. The recoveries of FB_1_ in the two spiked corn flour samples ranged from 97.5% to 99.23% and 94.54% to 99.25%, respectively. This method has the advantages of simple steps and low LOD, in comparison with the traditional ELISA, and also has the advantage of low cost, providomg an ideal means for rapid on-site detection of FB_1_ in samples.

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
