# Peer review of "Establishment of an Improved ELONA Method for Detecting Fumonisin B1 Based on Aptamers and Hemin-CDs Conjugates"

_sensors, 2022, doi:10.3390/s22176714_

Round 1

Reviewer 1 Report

I feel that you should emphasize, in the Discussion section, the fact that your - quite interesting - methodological approach requires considerable time (at least one hour)  to deliver results. In this respect, it should be more suitable for centralized laboratory used rather as an actual Point of Test biosensor. This is a relative disadvantage compared to electrochemical biosensors. Admittedly, there are limited responses on electrochemical biosensor development for the detection of fumonisin B1, however it cannot be excluded that such reports will soon appear. Usually, electrochemical biosensors allow for minute-fast and very sensitive (LOD 0,5-1 ng/ml) assays. An example is given by the following reference:

Chemosensors 20208, 121. https://doi.org/10.3390/chemosensors8040121

Therefore, I advise that you revise your Discussion by adressing this issue and make suggestions for future research.

Author Response

Point 1: I feel that you should emphasize, in the Discussion section, the fact that your - quite interesting - methodological approach requires considerable time (at least one hour) to deliver results. In this respect, it should be more suitable for centralized laboratory used rather as an actual Point of Test biosensor. This is a relative disadvantage compared to electrochemical biosensors. Admittedly, there are limited responses on electrochemical biosensor development for the detection of fumonisin B1, however it cannot be excluded that such reports will soon appear. Usually, electrochemical biosensors allow for minute-fast and very sensitive (LOD 0,5-1 ng/ml) assays. An example is given by the following reference: Chemosensors 2020, 8, 121. https://doi.org/10.3390/chemosensors8040121

Therefore, I advise that you revise your Discussion by adressing this issue and make suggestions for future research.

Response 1: Thank you for your suggestion, we have added a discussion section to the literature and made suggestions for future research. “The detection process of the method in this study requires at least 1 h, which is relatively long compared with that of the electrochemical biosensor. Therefore, more sensitive materials coupled with aptamers can be selected to establish a more rapid and sensitive detection method in future studies. And electrochemical methods can be combined with competitive immunoreactivity and aptamer-based assays where different particles (e.g. AuNPs, magnetic nanoparticles, microplates) are functionalized with aptamers for fast detection, high specificity and low LOD. The improved ELONA method established in this study can be used for the subsequent development of commercial kits to detect FB1 in food and feed like conventional ELISA kits after pre-treatment of enzyme-labeled plates with aptamer F10 and validation of its timeliness. The main problems of using current ELISA kits include high cost and narrow detection range, while the method established in this study can effectively circumvent these drawbacks.”

Reviewer 2 Report

General comment:

In this manuscript, authors claimed to establish a signal-off mechanism to detect Fumonisin B1 in corn flour. There are several issues and questions regarding the interpretation of data in the manuscript that authors need to address before it can be accepted. Some of the experiments should be repeated for consistent result. A major revision is recommended.

Specific comments:

1.    Abstract: At the end of abstract, authors mentioned the total detection time was 1 hour. However, figure 3E and line 295-300 in the main text suggested the time should be 150 minutes, which was way longer than 1 hour. Please explain.

2.  Figure 1: authors presented the particle size as the first figure. How did the particle size impact the sensing performance?

3.     Figure 4: Authors mentioned the experiment was performed with 10 ng/ml toxin, how did this number relate to the concentration in actual flour sample?

4.    Figure 4 and 5A: for the same concentration of 10 ng/ml FB1, Figure 5A showed about 60% higher absorbance. It indicated huge inconsistency between experiments. Please repeat experiment carefully before the manuscript can be accepted.

Author Response

Point 1: Abstract: At the end of abstract, authors mentioned the total detection time was 1 hour. However, figure 3E and line 295-300 in the main text suggested the time should be 150 minutes, which was way longer than 1 hour. Please explain.

Response 1: I'm sorry for the confusion, the detection time calculated in this study was prepared by pre-encapsulating the aptamer F10 of FB1 on the enzyme labeling plate. Then the samples and F10-com-Hemin-CDs couples were added and incubated at room temperature and protected from light for 30 min, washed three times with PBST, 100 μL TMB was added to each well and incubated at room temperature and protected from light for 30 min, and finally the reaction was terminated by adding 10% sulfuric acid solution and the absorbance value at 450 nm was measured, so the detection time was 1 h.

Point 2: Figure 1: authors presented the particle size as the first figure. How did the particle size impact the sensing performance?

Response 2: Thank you very much for your suggestion. Figure 1 shows the characterization of Hemin-CDs by TEM, and the results presented in the figure indicate that the prepared Hemin-CDs form a uniform diffuse spherical structure with an average particle size of 3.64 nm. The texture is uniform and diffuse, and can be used for subsequent experiments. The size of CDs was mostly less than 10 nm, and the smaller the particle diameter of CDs, the higher the toxicity, and the more toxic the aggregated CDs than the dispersed CDs. Particle size has little relationship with sensor performance.

Point 3: Figure 4: Authors mentioned the experiment was performed with 10 ng/ml toxin, how did this number relate to the concentration in actual flour sample?

Response 2: The concentration chosen for the specificity identification is based on a larger concentration within the detectable range of the method, and the same concentration is chosen for all targets. The amount of target added during the actual sample testing is based on the range specified in the national standard. We apologize for this. In the Materials and Methods, line 176 states that the concentration of each toxin is 100 ng/mL, but in the results, I am sorry to say that it is 10 ng/mL, we have changed it to 100 ng/mL.

Point 4: Figure 4 and 5A: for the same concentration of 10 ng/ml FB1, Figure 5A showed about 60% higher absorbance. It indicated huge inconsistency between experiments. Please repeat experiment carefully before the manuscript can be accepted.

Response 4: We apologize for this. In the Materials and Methods, line 176 states that the concentration of each toxin is 100 ng/mL, but in the results, we apologize for the 10 ng/mL, which we have revised to 100 ng/mL.

Round 2

Reviewer 2 Report

Thanks for the reply from authors! However, there are still confusions as below and authors need a lot of improvement on presentation...

1: I understand the one hour detection time that author explained. But it is still very confusing when figure 3 showed incubation time. If there could be supplementary file, figure 3 should be there. Still, authors need to present clearly.

2. There is no explanation in the main text describing the requirement on 10 nm diameter of the particle. It will still be confusing for readers.

3. Authors did not answer the question here... regardless of the typo on 10 ng/ml instead of 100 ng/ml as corrected, which was a really careless mistake and made me doubt on the credibility, the question was about the actual concentration of Fumonisin in real flour sample. Because from session 3.5, it looks like the detection was done by spiking in Fumonisin into flour sample. The number will allow readers to understand how sensitivtiy is this assay for real application. In addition to answering the question, authors should also explain in the main text.

Author Response

Point 1: I understand the one hour detection time that author explained. But it is still very confusing when figure 3 showed incubation time. If there could be supplementary file, figure 3 should be there. Still, authors need to present clearly.

Response 1: Thank you very much for your suggestion. We have included “F10 encapsulated on an enzyme labeling plate” in the supplemental material and indicated the timing of the assay in the Results and Discussion section.

Point 2: There is no explanation in the main text describing the requirement on 10 nm diameter of the particle. It will still be confusing for readers.

Response 2: Thanks for your reminder, we added "The size of CDs was mostly less than 10 nm" in lines 212-213 of the manuscript.

Point 3: Authors did not answer the question here... regardless of the typo on 10 ng/ml instead of 100 ng/ml as corrected, which was a really careless mistake and made me doubt on the credibility, the question was about the actual concentration of Fumonisin in real flour sample. Because from session 3.5, it looks like the detection was done by spiking in Fumonisin into flour sample. The number will allow readers to understand how sensitivtiy is this assay for real application. In addition to answering the question, authors should also explain in the main text.

Response 3: Thank you very much for your careful review and we apologize for our carelessness, again. The corn flour used in the experiment was purchased from a local supermarket as an edible raw material and the laboratory test showed no background values. In order to verify the practicality of the method in real samples. Different concentrations of FB1 were added to the pretreated samples.

We added the criteria for FB1 content in edible corn flour in the introduction section of the manuscript, and added in the results and discussion section that the method meets the national requirements for the determination of FB1 content. “Since the actual samples were tested with 1 g of corn sample dissolved in 5 mL of solution, the limit values of 800 μg/kg and 200 μg/kg for FB1 in corn flour can be converted. With 200 μg/kg as the calibration value, 1 kg of corn flour was dissolved in 5 L solution, the upper limit of FB1 was converted to 40 ng/mL, and the LOD of this method was 4.30 ng/mL. Therefore, this method meets the detection requirements of the standard.”
